# Risk Perception and Fatigue in Port Workers: A Pilot Study

**DOI:** 10.3390/ijerph21030338

**Published:** 2024-03-13

**Authors:** Clarice Alves Bonow, Valdecir Zavarese da Costa, Leticia Silveira Cardoso, Rita Maria Heck, Jordana Cezar Vaz, Cynthia Fontella Sant’Anna, Julia Torres Cavalheiro, Gabriela Laudares Albuquerque de Oliveira, Thaynan Silveira Cabral, Carlos Henrique Cardona Nery, Mara Regina Santos da Silva, Marta Regina Cezar-Vaz

**Affiliations:** 1Faculty of Nursing, Federal University of Pelotas, Pelotas 96010-610, Brazil; rmheckpillon@yahoo.com.br (R.M.H.); julia.cavalheiro@ufpel.edu.br (J.T.C.); gabriela.laudares@ufpel.edu.br (G.L.A.d.O.); mrcezarvaz@furg.br (M.R.C.-V.); 2Department of Nursing, Federal University of Santa Maria, Santa Maria 97105-900, Brazil; valdecir.costa@ufsm.br (V.Z.d.C.); thaynan.cabral@acad.ufsm.br (T.S.C.); 3Department of Nursing, Federal University of Pampa, Uruguaiana 97501-970, Brazil; leticiacardoso@unipampa.edu.br (L.S.C.); cynthiasantanna@unipampa.edu.br (C.F.S.); 4Institute of Dermatology Professor Rubem David Azulalay (Medical Residency), Rio de Janeiro 20020-020, Brazil; jordana.cezarvaz@gmail.com; 5Institute of Human and Information Sciences—ICHI, Santa Vitória do Palmar Campus, Federal University of Rio Grande, Santa Vitória do Palmar 96230-000, Brazil; carloscardona@furg.br; 6School of Nursing, Federal University of Rio Grande, Rio Grande 96203-900, Brazil; denfmara@furg.br

**Keywords:** risk perception, fatigue, work, ports, occupational health

## Abstract

Introduction and Objectives: The aims of this study were to assess fatigue in port workers; analyze the association between fatigue and levels of trust in organizations, as well as the association between authorities and risk perception; and examine the official documents governing the studied port, along with the current health and communication status of the port workers. Materials and Methods: This was a descriptive and cross-sectional pilot study, which presented quantitative and qualitative data, and it was carried out among port workers in the city of Pelotas, Rio Grande do Sul, Brazil. Thirty-nine port workers responded to quantitative questionnaires, which collected their socio-demographic data, as well as a risk perception questionnaire, the Chalder Fatigue Scale, and the Checklist of Individual Strength. Five documents from the port regiment were studied and qualitatively analyzed. The health communications consisted of presenting infographics with research data and providing information for reducing fatigue. Results: Fifteen workers (38.5%) were considered fatigued. There was a reduction in fatigue associated with trust in the unions and the labor management body, and there was an agreement that the precarious environment was completely unacceptable. The qualitative data in the documents indicated that it was possible to identify the infrastructure of the port environment, the legislation, the strategies to be adopted in cases of natural disasters, emergency plans, plans for the protection and promotion of workers’ health, individual and collective protection plans, the division of the sectors and those responsible for them, and documents detailing the hierarchy within the ports. The qualitative analysis culminated in graphic representations (infographics) created to communicate the research results to port workers, specifically in relation to fatigue, and we presented the ways to prevent fatigue at work. Discussion/Limitations: Studying the risk perceptions and fatigue levels of port workers through research with the active participation of these workers presented their lived experiences, which promoted discussion and perhaps more effective proposals to change their work conditions.

## 1. Introduction

Considering the relevance of the port environment for global development, it is important to investigate not only the environment/work and the port’s structure, for example, but also the workers who carry out the work because by understanding the specific challenges of this work, it is possible to propose improvements to directly contribute to the health and safety of workers.

Different studies among port workers have presented the development of work-related health conditions [1,2,3]. A study carried out on 290 dockworkers in the Philippines sought to determine the prevalence and causes of work-related musculoskeletal disorders. The results presented indicated more important symptoms in the workers’ upper and lower back regions, in addition to a positive association between the length of work experience and the severity of musculoskeletal symptoms [1]. Another investigation analyzed the outpatient care provided by a health team to port workers at a port in Costa Rica. Many cases of drug use were identified, as well as musculoskeletal problems and headaches [2]. Exposure to benzene among port workers in southern Italy was presented in one study. An evaluation of workers’ urine samples identified a biological marker for benzene, which was excreted in greater quantities at the end of the work shift [3].

Due to the socio-environmental risks of port work, port workers move cargo and carry out other work activities in a dangerous and unhealthy environment, with the risks increasing in correlation with the cargo being handled, exposing the workers to permanent health dangers.

Being in such an environment means exposing oneself to working conditions, which create health risks. Risk is associated with an adverse event, activity, or physical attribute with certain objective probabilities of causing damage, and it can be estimated through different methods, such as statistical predictions, estimation probabilistic risk analyses, risk/benefit comparisons, or psychometric analyses, which calculate acceptability levels in order to allow for standards to be established [4].

Considering the risks mentioned, the state of fatigue in a port worker can be an aggravating factor in increasing the risks already established in the environment, as a fatigued worker has a decreased ability to focus and pay attention than one who does not present fatigue. Fatigue reduces performance, increases absenteeism, and contributes to the development of musculoskeletal disorders [5].

Fatigue is understood as an overwhelming feeling of exhaustion and decreased physical and mental work capacities [6]. In work environments, concerns over fatigue are related to increasing worker productivity and safety. The literature presents studies in the fields of aviation [7], nursing [8], driving [9], and construction [10].

To establish effective risk communication—particularly regarding fatigue among workers handling chemical substances, transporting and storing dangerous materials and loads, managing toxic and radioactive products, and handling heavy loads [11], such as port workers—this pilot study is conducted with the aim of assessing fatigue in port workers; analyzing the association between fatigue and levels of trust in organizations, as well as the association between authorities and risk perception; and examining the official documents governing the studied port, along with the current health and communication status of the port workers.

## 2. Materials and Methods

### 2.1. The Context of the Pilot Study

This pilot study was part of a larger research project involving port workers at a seaport in the south of Brazil. The research team has been carrying out studies on the health and safety of port workers since 2006 [12]. The proposal included studying other ports and structured and semi-structured interviews with workers; however, after the quantitative data were collected by the group of workers in January and February 2020, the COVID-19 pandemic began. In this context, face-to-face activities were suspended, requiring activities to be reorganized remotely. With the progress of the pandemic and the difficulties in returning to face-to-face activities, the workers interviewed were dismissed, losing their connection with the institution and their access to the institutional email where they received messages from the research team. Therefore, the continuation of the project was made unfeasible.

### 2.2. Research Design and Setting

This was a descriptive and cross-sectional study presenting quantitative and qualitative data, which complemented each other in the analysis of the research object. The study was carried out among port workers in the municipality of Pelotas, Rio Grande do Sul, Brazil (Appendix A). For this study, structured interviews and document analyses were carried out. The Research Ethics Committee of the Faculty of Medicine of the Federal University of Pelotas approved the study (CAAE 26989919.9.0000.5317). All participants signed the study consent form.

The municipality of Pelotas is located in the south of the State of Rio Grande do Sul (Appendix A), Brazil, on the banks of the São Gonçalo channel, which connects the Patos and Mirim lagoon. This port plays an important role in the economic development of the southern half of Brazil, generating work and income and reducing logistics costs for importing and exporting companies in the region [13].

The movements in the Port of Pelotas totaled 1.3 million tons in 2021 [14]. The cargo transported at the port includes rice and other bulk products, frozen meat and chicken, fruit, mineral coal, petroleum derivatives, clay, salt, fertilizer, agricultural machinery, and wooden logs (Appendix A) [15].

### 2.3. Participants

For the quantitative stage, the study population comprised 44 workers from the port of Pelotas, Rio Grande do Sul, Brazil. All workers were associated with the workforce management agencies, which are responsible for the number of workers necessary to carry out the work and manage and supply port labor, control the rotation of workers, pay remuneration, collect charges, and ensure compliance with health and safety standards [16]. In total, 5 workers refused to participate in the study; thus, 39 workers participated in the research.

Following the qualitative stage, official documents available on the management website of the studied port were analyzed (https://www.portosrs.com.br/site/comunidade_portuaria/pelotas/conheca_o_porto—accessed on 9 March 2024).

### 2.4. Questionnaires and Data Collection

Data collection was carried out from January to February 2020 during a shift arranged with the port management company and in the participants’ own work environment to minimally interrupt their work routines. Four questionnaires were used. One questionnaire was socio-demographic and included questions regarding age, sex, marital status, skin color, education, income, and length of experience; there was also an adapted risk perception questionnaire [17]. In this study, questions were asked about the risks perceived in the work environment and the trust of port workers regarding organizations/authorities related to the risks in the work environment, and requests were made for individual and collective risk assessments. Furthermore, the following two questionnaires were used to assess fatigue: the Chalder Fatigue Scale and the Checklist of Individual Strength (Portuguese version). The Chalder Fatigue Scale has British origins and is widely used to assess physical and mental fatigue. Formulated by Chalder and his collaborators in 1993 [18], this scale was translated and adapted into Brazilian Portuguese and validated by Hyong Jin Cho and his collaborators in 2007 [19]. The questionnaire has 11 items and contains questions regarding the symptomatology of physical and mental fatigue. It is a Likert-type questionnaire, scoring each symptom of fatigue measured from zero to three and relating their intensity, and its calculation takes place using a bimodal score. In this calculation, considering this type of scale, values between zero and one are considered zero, and values of two and three are considered as one, with a sum of the values greater than or equal to four being characterized as fatigue [19].

The Checklist of Individual Strength (CIS) was developed in 1994 [20] and validated in 2013 [21]. It consists of 20 multidimensional items, and it assesses fatigue dimensions and reveals psychological fatigue measurement characteristics. This questionnaire covers the following four dimensions: the subjectivity element of fatigue or subjective fatigue, decreases in motivation, reductions in physical activity, and reductions in concentration. It can measure the severity of fatigue [20,21]. For this study, the results for subjective fatigue (CIS8R) were selected, as this is the most often reported result [22], thus bringing our possible results closer to those in the literature. The score was obtained from the results of the sum of the questions. For the qualitative stage, official documents from the Port of Pelotas were accessed, and they are available online on the government’s website.

### 2.5. Data Analysis

Quantitative variables were described using means and standard deviations (symmetric distributions) or medians and interquartile ranges (asymmetric distributions), depending on the distribution of the variables. The decision for the type of distribution was based on the Shapiro–Wilk normality test. Categorical variables were described using absolute and relative frequencies.

To compare the means, Student’s t-test was applied. This parametric test compares the means of two distinct populations. In the case of asymmetry, the Mann–Wallis test was used. This non-parametric test compares the medians of two independent populations.

To evaluate the association between categorical variables, Pearson’s chi-square or Fisher’s exact tests were applied. The chi-square is a non-parametric test used to evaluate a discrepancy between a set of observed frequencies and another set of expected frequencies, according to the hypothesis of independence between the variables (the null hypothesis of the test). Fisher’s exact test replaces the chi-square test when 25% or more of the cells have a frequency lower than 5.

The agreement between the scales assessing fatigue was assessed using the kappa coefficient. The closer the score was to one, the greater the agreement between the scales.

To control for confounding factors, a Poisson regression analysis was used. The criteria for entering the variable into the model were that it presented a *p*-value of <0.20 in the bivariate analysis, and the criterion for its permanence in the model was that it presented a *p*-value of <0.10 in the final model.

The significance level adopted was 5% (a *p*-value of ≤0.05), and the analyses were carried out using SPSS version 21.03.

The qualitative stage underwent content analysis [23]. For this stage, after accessing the official documents, a reading and subsequent search were carried out using the following keywords: worker, port, health, disease, and risk. In this way, it was possible to objectively identify how the port authorities included these topics in the organization of the service. An analysis of the frequency of the search terms was carried out. After the floating reading, a vertical analysis of the data was carried out, establishing the categories and subcategories.

## 3. Results

### 3.1. Quantitative Results

The study population comprised 39 workers from the port of Pelotas, resulting in a participation rate of 88.6%. The participants’ average age was 46.6 years (± 10.6), and they were predominantly male (84.6%), married/with a partner (56.4%), and white (87.2%), with a maximum education of up to high-school level (56.4%) and a monthly income of three times the minimum wage. The mean average of years of education was 12.4 years (± 2.6). The median time working in the sector was 9 years (25–75 percentiles: 8–12), and only two workers had another paid activity (5.1%), as shown in Table 1.

Table 2 presents the results regarding workers’ trust in the organizations and authorities when referring to risks associated with their work environment. Considering the maximum level of seven points, good levels of trust in the organizations/authorities were observed, as the minimum average score was five points. The highlights included higher levels of trust in the port guard (average of 6.8) and unions (average of 6.4) and lower levels of trust in the customs brokers (average of 5) and labor management bodies (average of 5.3).

Data on the precarious work environment and its risks are presented in Table 3. The highest averages indicated that workers believed that a precarious work environment would pose greater health risks and that exposure to one of the risks could have consequences for the rest of their lives. It is important to note that all port work is connected to workforce management agencies, as they are responsible for overseeing the entire process of port work and promoting the training and professional development of port workers.

Table 4 presents the results of the perceptions of individual risks of port workers, with workers perceiving higher levels of risk in relation to noise (median of six) and fire and air pollution (median of four for both).

Table 5 presents a comparison between the risk perceptions of workers at the port of Pelotas individually and for people in general. Workers perceived significantly higher risks of traffic accidents (*p* <0.001) and global warming (*p* = 0.033), and the total risk score (*p* = 0.007) for people in general was higher than the risk for themselves.

Table 6 presents the results of the scales assessing fatigue. For the Chalder Fatigue Scale, 30.8% (*n* = 12) of port workers were considered fatigued. It is important to highlight that due to the greater number of physical than mental items with the same median, the fatigue appeared to be more mental than physical. In relation to the CIS, six workers (15.4%) were considered fatigued. The value of the kappa coefficient evaluating the agreement between the two scales (Chalder and CIS) was 0.16 (*p* = 0.267), which was considered low intensity; therefore, there was no good agreement between the scales.

Considering the presence of fatigue as a score above the cut-off point for either scale, fifteen workers (38.5%) were considered fatigued. Its associations with the socio-demographic and work variables are presented in Table 7. There was no statistically significant association with any variable.

Table 8 presents the association of fatigue with the levels of confidence and risk perception. Trust in the unions was significantly lower among workers who presented with fatigue (*p* = 0.045).

After controlling for confounding factors, the levels of trust in unions (*p* < 0.001) and labor management bodies (*p* < 0.001) and the agreement that a precarious work environment was significantly associated with fatigue remained significantly associated with fatigue being completely unacceptable (*p* < 0.001) (Table 9). Workers scoring one point more for trust in unions had a 32% reduction in the probability of fatigue (RP = 0.68; 95% CI: 0.58–0.81). Those who had a higher level of trust in workforce management agencies also showed a 36% reduction in the occurrence of fatigue (RP = 0.64; 95% CI: 0.49–0.83). Finally, workers with an additional point for the agreement that a precarious environment was completely unacceptable presented a reduction in the probability of fatigue of 29% (RP = 0.71; 95% CI: 0.58–0.87).

### 3.2. Qualitative Results

The official documents structuring the port planning with the aim of establishing guidelines for the development of Brazilian port complexes were analyzed. Five documents referring to the port studied were identified [24,25,26,27,28]. In the documents, it was possible to identify the entire infrastructure of the port environment, the legislation, and the strategies to be adopted in cases of natural catastrophes, as well as emergency plans, plans for the protection and promotion of workers’ health, individual and collective protection plans, division of the sectors and those responsible for them, and documents detailing the hierarchy within the ports, among others. Searching for words within the documents, we sought to identify the most or least prevalent themes. The frequencies of the words searched are shown in Table 10.

The following two categories were identified in the analysis of the organizational documents of the Port of Pelotas: infrastructure and organization and risk prevention. The documents generally presented the definitions, plans, and standards, which ports must follow (Table 11).

The documents analyzed presented the current national standards for organizing work in general, with mention of the Environmental Risk Prevention Program (PPRA) [29] and Occupational Health Medical Control Program (PCMSO) [30]. The two programs are part of a set of Occupational Health and Safety Regulatory Standards set by the Ministry of Health.

The PPRA aims to preserve the physical integrity and health of workers and the environment, anticipating and evaluating potential risks, which may exist in the work environment [29].

The PCMSO is a program carried out annually, presenting a list of all positions in the company and their respective risk factors at work. For this, periodic exams are defined for workers [30]. Furthermore, the documents mentioned NR29 [31], which is specific to port workers and addresses actions directly related to the risks and situations involving port work and helps to develop safety measures aimed at preventing workers from being affected by occupational pathologies and work accidents [31].

The qualitative analysis culminated in graphic representations (infographics) created to inform port workers about the research results, specifically in relation to fatigue (Figure 1), and they presented the ways to prevent fatigue at work (Figure 2). The infographics were sent to the institutional email addresses of the 44 port workers, asking them whether the instructions could be followed within the work environment. However, there were no responses from them regarding the materials sent, as the workers lost access to their institutional email addresses after the port was shut down during the pandemic.

## 4. Discussion

The results presented revealed important aspects regarding the demographics, risk perceptions, fatigue, and organizational trust among port workers. The predominance of male workers indicated a common characteristic in industrial sectors, including port operations [1]. This gender distribution can influence the social dynamics in a workplace, and it requires specific approaches to occupational health issues, considering the specific needs of male workers.

Another important socio-demographic data point is that many port workers (56.4%) had education up to high-school level. This data point highlighted the possible need for training and continuing education programs to improve job-specific skills and promote professional advancement. It could also guide how interventions should be carried out in this group, indicating the need for more objective and practical and less theoretical propositions. Furthermore, one could reflect on the documents qualitatively analyzed in this study. They are available on the website of the port studied, but they are written in formal language, which makes access difficult for people with less education.

Regarding trust in organizations/authorities, the minimum average of five points suggested that, in general, workers had trust in organizations and authorities in relation to their safety at work. This trust is crucial for the effective implementation of occupational safety practices and policies, such as the infographics presented in this study.

Trust is a key component in risk communication [35,36,37]. It is estimated that information regarding managers’ efforts and regulatory rules to prevent employee discharge may be more important in most “pre-accident” situations, as they may be the main place of trust for the public [38]. In the case of the present study, the documents analyzed in the qualitative stage indicated that the materials were old, but they also indicated the need for the work to be carried out only during the day shift, which may have justified the greater trust in managers.

In contrast, a study carried out during the pandemic on public trust in authorities in Singapore [39] showed that trust was recognized as a crucial component of effective risk management. However, in dangerous work, such as port work, and in a delicate moment, such as the pandemic, it is understood that public trust based on perceptions of a manager’s competence and care can lead to underestimating risks, and thus, this can reduce one’s belief in the need to take individual measures to control risks [39].

However, the high perception that a precarious work environment would pose greater health risks may have indicated that port workers recognized the importance of safe working conditions. A study carried out in China among port workers emphasized the physical, chemical, and psychological risks they were exposed to, highlighting the importance of putting effective prevention measures into practice to guarantee the health of port workers [40]. Furthermore, the perceptions of individual risks in relation to noise, fire, and air pollution may have indicated specific areas requiring priority attention in terms of the control and prevention of occupational risks in this environment.

Another important issue is the perception of individual and collective risk. Port workers perceived a significantly higher risk of traffic accidents and global warming, and there was a higher total risk score for the collective than for the individual. This difference revealed that workers may have felt less vulnerable to certain risks, which highlighted the importance of personalized and focused awareness strategies.

The identification of a significant percentage of workers with fatigue (38.5%) highlighted the importance of managing the mental and physical health of these professionals. The application of the Chalder Fatigue Scale in a study carried out in Japan [41] among administrative and non-administrative workers showed that there was no statistically significant difference between the two types of workers. The observation that fatigue appeared to be more mental than physical was justified. This result may have indicated that psychological aspects, such as mental fatigue, should be prioritized as intervention initiatives for this group of workers.

The correlation between lower trust in the unions and higher incidence of fatigue suggested the importance of effective support and representation for these organizations in defending workers’ interests. The same correlation applied to trust in workforce management agencies. A study carried out by teleworking workers investigated the relationship between teleworking and fatigue, demonstrating that fatigue was the factor, which most negatively affected confidence [42]. Although the work was different, these findings presented an important trend, which should be clarified.

Furthermore, the relationship between the perception that a precarious environment is completely unacceptable and a lower probability of fatigue highlighted the positive impact that the perception of adequate working conditions can have on the mental and physical health of port workers. Health and work have an inseparable relationship, which is why it is essential to think about healthy work [43].

## 5. Conclusions

By studying the risk perceptions and fatigue levels of port workers, including their active participation in the research, we can present the reality of their experience of this work. This can promote discussion and perhaps more effective proposals to improve work conditions. Including workers’ perspectives in the research process not only validates their experiences but also provides valuable information, which may not be fully understood by outside observers. Specifically in relation to fatigue, this is important because although many of the workers interviewed were not fatigued, this did not mean that their work was not tiring. The experiences of port workers can vary widely depending on the type of cargo being handled, weather conditions, the equipment used, and other factors specific to the workplace. The active participation of workers helps in contextualizing the proposed solutions, ensuring that they are adapted to the unique conditions of each port. We suggest carrying out more studies in different locations to broaden this discussion and propose the strategies to improve port work.

## Figures and Tables

**Figure 1 ijerph-21-00338-f001:**
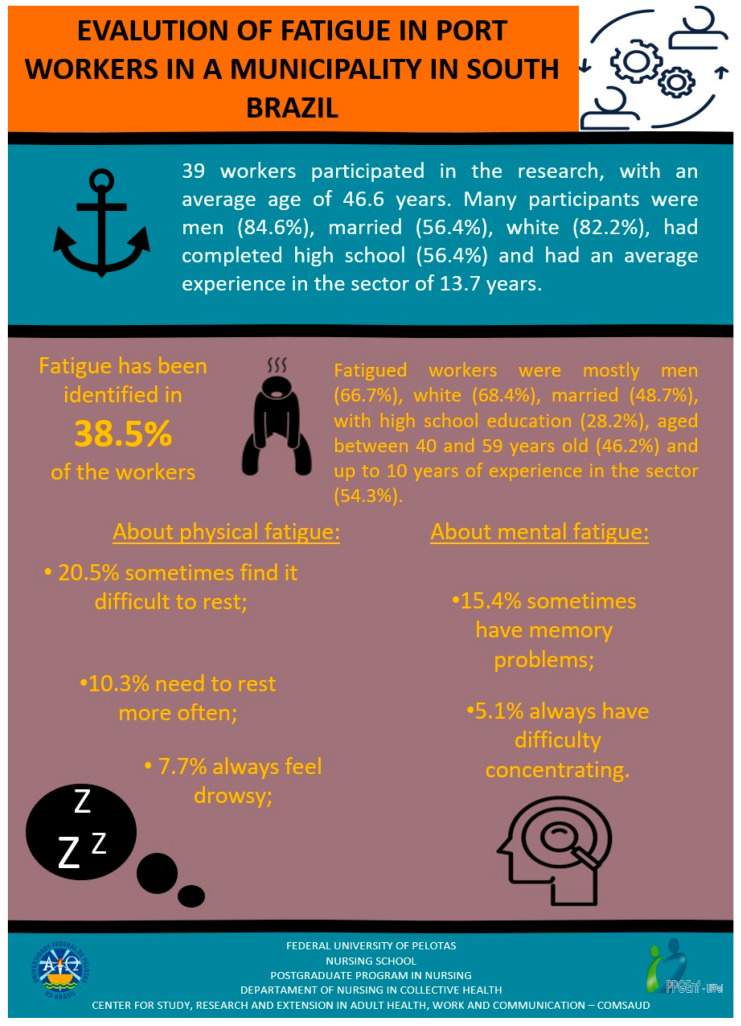
Infographic 1 presented to port workers.

**Figure 2 ijerph-21-00338-f002:**
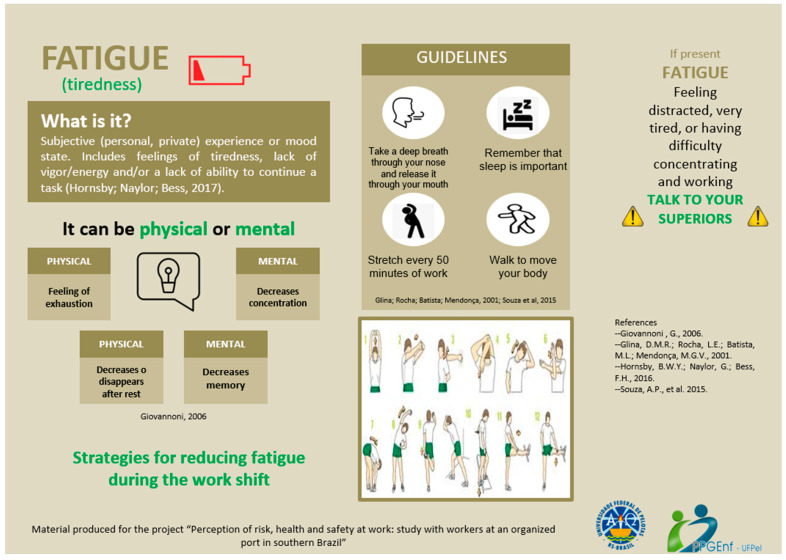
Infographic 2 presented to port workers [6,32,33,34].

**Table 1 ijerph-21-00338-t001:** Population characteristics.

Variable	*n* = 39
Age (years), average ± SD	46.6 ± 10.6
Sex, *n* (%)	
Male	33 (84.6)
Female	6 (15.4)
Marital status, *n* (%)	
Single/no partner	17 (43.6)
Married/with partner	22 (56.4)
Race, *n* (%)	
White	34 (87.2)
Afro-descendent	5 (12.8)
Educational level, *n* (%)	
Up to high school	22 (56.4)
Higher education or more	17 (43.6)
Educational level (years), average ± SD	12.4 ± 2.6
Years spent in port work (years), median (P25–P75)	9 (8–12)
Income (minimum wage) **, *n* (%)	
Up to 3	17 (43.6)
More than 3	22 (56.4)
Other paid activity, *n* (%)	
No	37 (94.9)
Yes *	2 (5.1)

* Self-employed and entrepreneur; ** The minimum wage is BRL 1265.63, equivalent to USD 5416.

**Table 2 ijerph-21-00338-t002:** Data on workers’ levels of trust in the organizations and authorities when referring to risks associated with their work environment.

Organization/Authority	Variation	Average ± SD	Min–Max
Port Superintendent	1–7	5.8 ± 0.8	4–7
Port Authority Board	1–7	5.5 ± 0.9	3–7
Unions	1–7	6.4 ± 1.2	1–7
Port Operators	1–7	5.5 ± 1.1	2–7
Customs Brokers	1–7	5.0 ± 1.4	2–7
Port Guard	1–7	6.8 ± 0.6	4–7
Workforce Management Agencies	1–7	5.3 ± 1.5	2–7
Port Work Accident Prevention Commission	1–7	5.8 ± 0.7	5–7
Specialized Service in Safety and Health at Port Work	1–7	5.5 ± 0.9	4–7
Total Score (Average of the responses)	1–7	5.8 ± 0.5	4.1–6.6

**Table 3 ijerph-21-00338-t003:** Data on the precarious work environment and its risks.

Item	Variation	Average ± SD	Min–Max
A precarious work environment would pose risks (chemical, physical, biological, ergonomic/psychosocial, and/or mechanical/accidents) to health.	1–7	6.9 ± 0.4	5–7
If exposed to one of these risks, it could have consequences for the rest of my life.	1–7	6.7 ± 0.5	5–7
I would stop working there for fear of these risks.	1–7	2.6 ± 1.8	1–7
It would be completely unacceptable.	1–7	5.0 ± 1.6	1–7
It would be shameful.	1–7	4.8 ± 1.9	1–7
You would be exposed to chemical risks (involving exposure to chemical agents, such as alcohol, gasoline, grease, and solvents).	1–7	4.5 ± 1.8	1–7
You would be exposed to physical risks (such as noise and dust).	1–7	5.3 ± 1.4	1–7
You would be exposed to biological risks (such as viruses, bacteria, and parasites).	1–7	4.0 ± 1.8	1–6
You would be exposed to ergonomic/psychosocial risks (related to lifting and moving weights, inadequate working postures, repetitive movement, and excessive working hours).	1–7	5.0 ± 1.6	1–7
You would be exposed to mechanical risks/accidents (movement and handling of equipment in operation, falling equipment, tools, and materials, and the probability of explosions and fire).	1–7	4.9 ± 1.6	1–7
It would cause health problems for all workers.	1–7	6.6 ± 0.8	1–7
Total score (Average of the responses)	1–7	5.1 ± 0.6	3.0–6.4

**Table 4 ijerph-21-00338-t004:** Data on individual risk perceptions.

Item	Variation	Median (P25–P75)	Min–Max
Traffic accident	1–7	2 (1–3)	1–5
Domestic accident	1–7	2 (1–3)	1–7
Alcohol	1–7	1 (1–2)	1–6
Global warming	1–7	3 (2–4)	1–7
Genetically modified food	1–7	1 (1–1)	1–5
Noise	1–7	6 (5–7)	1–7
Natural catastrophe	1–7	3 (1–4)	1–7
Cigarettes	1–7	1.5 (1–3)	1–7
Ground contamination	1–7	3 (1–4)	1–7
Inappropriate diet	1–7	1 (1–3)	1–7
Drugs	1–7	1 (1–1)	1–7
Fire	1–7	4 (3–6)	1–7
Chemical industry	1–7	1 (1–1)	1–7
Obesity	1–7	2 (1–3)	1–6
Air pollution	1–7	4 (2–5)	1–7
Medical radiography	1–7	1 (1–1)	1–2
Chemical waste	1–7	1 (1–1)	1–3
Radioactive waste	1–7	1 (1–1)	1–2
Terrorism	1–7	1 (1–1)	1–2
Nuclear plant	1–7	1 (1–1)	1–2
Total score * (average of the responses)	1–7	2.3 ± 0.4	1.5–3.3

*, Described by means ± standard deviations.

**Table 5 ijerph-21-00338-t005:** Comparison between individual and collective risk perceptions.

Item	Individual Risks	Collective Risks	*p* ^a^
	Median (P25–P75)	Median (P25–P75)	
Traffic accident	2 (1–3)	4 (3–5)	<0.001
Domestic accident	2 (1–3)	3 (1–4)	0.118
Alcohol	1 (1–2)	2 (1–4)	0.072
Global warming	3 (2–4)	4 (3–5)	0.033
Genetically modified food	1 (1–1)	1 (1–1)	0.357
Noise	6 (5–7)	6 (4–6)	0.341
Natural catastrophe	3 (1–4)	3 (1–4)	0.379
Cigarettes	1.5 (1–3)	3 (2–4)	0.154
Ground contamination	3 (1–4)	3 (1–4)	0.633
Inappropriate diet	1 (1–3)	3 (1–4)	0.179
Drugs	1 (1–1)	1 (1–2)	0.241
Fire	4 (3–6)	5 (4–6)	0.367
Chemical industry	1 (1–1)	1 (1–1)	0.863
Obesity	2 (1–3)	2 (1–3)	0.066
Air pollution	4 (2–5)	4 (2–5)	0.706
Medical radiography	1 (1–1)	1 (1–1)	0.414
Chemical waste	1 (1–1)	1 (1–1)	0.368
Radioactive waste	1 (1–1)	1 (1–1)	0.655
Terrorism	1 (1–1)	1 (1–1)	0.317
Nuclear plant	1 (1–1)	1 (1–1)	0.317
Total score * (average of the responses)	2.3 ± 0.4	2.5 ± 0.6	0.007 ^b^

*, Described by means ± standard deviations; ^a^, Mann–Whitney test; ^b^, Student’s *t*-test.

**Table 6 ijerph-21-00338-t006:** Assessment of fatigue using the Chalder Fatigue Scale and the Checklist of Individual Strength (CIS).

Scale	*n* = 39
Chalder Fatigue Scale, median (P25–P75)	
Physical score (0–7 points)	1 (0–2)
Mental score (0–4 points)	1 (0–1)
Total score (0–11 points)	2 (0–4)
Chalder Fatigue Scale classification, *n* (%)	
No fatigue (<4 points)	27 (69.2)
With fatigue (≥4 points)	12 (30.8)
Checklist of Individual Strength (CIS), average ± SD	
Severe fatigue (8–56 points)	20.4 ± 9.5
Classification for CIS for severe fatigue, *n* (%)	
No fatigue (<35 points)	33 (84.6)
With fatigue (≥35 points)	6 (15.4)

**Table 7 ijerph-21-00338-t007:** Association of the socio-demographic and work variables with fatigue.

Variable	No Fatigue (*n* = 24)	With Fatigue (*n* = 15)	*p*
Age (years), average ± SD	48.4 ± 10.0	43.7 ± 11.3	0.178 ^a^
Sex, *n* (%)			0.180 ^b^
Male	22 (91.7)	11 (73.3)	
Female	2 (8.3)	4 (26.7)	
Marital status, *n* (%)			0.491 ^c^
Single/no partner	12 (50.0)	5 (33.3)	
Married/with partner	12 (50.0)	10 (66.7)	
Race, *n* (%)			0.631 ^b^
White	20 (83.3)	14 (93.3)	
Afro-descendent	4 (16.7)	1 (6.7)	
Educational level, *n* (%)			0.193 ^c^
Up to high school	16 (66.7)	6 (40.0)	
Higher education or more	8 (33.3)	9 (60.0)	
Educational level (years), average ± SD	11.9 ± 2.4	13.3 ± 2.7	0.106 ^a^
Years spent in port work (years), median (P25–P75)	9 (8–12)	10 (9–24)	0.482 ^d^
Income (minimum wage), *n* (%)			0.980 ^c^
Up to 3	11 (45.8)	6 (40.0)	
More than 3	13 (54.2)	9 (60.0)	
Other paid activity, *n* (%)			1.000 ^b^
No	23 (95.8)	14 (93.3)	
Yes	1 (4.2)	1 (6.7)	

^a^, Student’s *t*-test; ^b^, Fisher’s exact test; ^c^, Pearson’s chi-square test; ^d^, Mann–Whitney test.

**Table 8 ijerph-21-00338-t008:** Association between the levels of trust in organizations/authorities, the precarious environment, and risks and levels of risk perception with fatigue.

Variable	No Fatigue (*n* = 24)	With Fatigue (*n* = 15)	*p*
Levels of trust in organizations/authorities			
Port Superintendent	5.7 ± 0.9	5.9 ± 0.7	0.575 ^a^
Port Authority Board	5.5 ± 1.0	5.6 ± 0.8	0.854 ^a^
Unions	6.7 ± 0.6	5.9 ± 1.7	0.045 ^a^
Port Operators	5.6 ± 1.2	5.4 ± 0.9	0.675 ^a^
Customs Brokers	4.7 ± 1.3	5.6 ± 1.3	0.135 ^a^
Port Guard	6.9 ± 0.3	6.6 ± 0.9	0.119 ^a^
Workforce Management Agencies	5.6 ± 1.4	4.8 ± 1.5	0.096 ^a^
Port Work Accident Prevention Commission	5.8 ± 0.6	5.8 ± 0.9	0.962 ^a^
Specialized Service in Safety and Health at Port Work	5.4 ± 0.9	6.0 ± 0.8	0.230 ^a^
Total Score (Average of the responses)	5.8 ± 0.5	5.7 ± 0.5	0.430 ^a^
Precarious work environment and its risks			
A precarious work environment would pose risks (chemical, physical, biological, ergonomic/psychosocial, and/or mechanical/accidents) to my health.	6.9 ± 0.3	6.8 ± 0.6	0.393 ^a^
If exposed to one of these risks, it could have consequences for the rest of my life.	6.7 ± 0.6	6.7 ± 0.5	0.884 ^a^
I would stop working there for fear of these risks.	2.4 ± 1.8	2.9 ± 1.9	0.402 ^a^
It would be completely unacceptable.	5.4 ± 1.6	4.3 ± 1.3	0.027 ^a^
It would be shameful.	5.3 ± 1.6	3.9 ± 2.1	0.024 ^a^
You would be exposed to chemical risks (involving exposure to chemical agents, such as alcohol, gasoline, grease, and solvents).	4.5 ± 1.8	4.6 ± 1.7	0.812 ^a^
You would be exposed to physical risks (such as noise and dust).	5.5 ± 1.2	4.9 ± 1.6	0.222 ^a^
You would be exposed to biological risks (such as viruses, bacteria, and parasites).	4.0 ± 1.7	3.9 ± 1.9	0.855 ^a^
You would be exposed to ergonomic/psychosocial risks (related to lifting and moving weights, inadequate working postures, repetitive movement, and excessive working hours).	4.9 ± 1.7	5.2 ± 1.3	0.593 ^a^
You would be exposed to mechanical risks/accidents (movement and handling of equipment in operation, falling equipment, tools, and materials, and probability of explosions and fire).	4.9 ± 1.7	4.9 ± 1.5	0.927 ^a^
It would cause health problems for all workers.	6.7 ± 0.6	6.3 ± 1.1	0.240 ^a^
Total Score	5.2 ± 0.6	4.9 ± 0.7	0.220 ^a^
Risk Perception Levels			
Traffic accident	2 (1–3)	2 (1–3)	0.638 ^b^
Domestic accident	2 (1–4)	2 (1–3)	0.853 ^b^
Alcohol	1.5 (1–2)	1 (1–3)	0.853 ^b^
Global warming	3.5 (2–5)	3 (1–4)	0.191 ^b^
Genetically modified food	1 (1–1)	1 (1–2)	0.809 ^b^
Noise	6 (5–7)	5 (4–6)	0.146 ^b^
Natural catastrophe	3 (1–5)	4 (1–4)	0.721 ^b^
Cigarettes	1.5 (1–4)	1.5 (1–3)	0.709 ^b^
Ground contamination	3 (1–5)	2 (1–3)	0.202 ^b^
Inappropriate diet	1 (1–3)	3 (1–3)	0.323 ^b^
Drugs	1 (1–1)	1 (1–1)	0.687 ^b^
Fire	4.5 (3–6)	4 (3–6)	0.966 ^b^
Chemical industry	1 (1–1)	1 (1–2)	0.875 ^b^
Obesity	1 (1–3)	2 (1–3)	0.484 ^b^
Air pollution	4 (3–5)	3 (2–4)	0.103 ^b^
Medical radiography	1 (1–1)	1 (1–1)	0.831 ^b^
Chemical waste	1 (1–1)	1 (1–1)	0.721 ^b^
Radioactive waste	1 (1–1)	1 (1–1)	0.831 ^b^
Terrorism	1 (1–1)	1 (1–1)	0.831 ^b^
Nuclear plant	1 (1–1)	1 (1–1)	0.831 ^b^
Total score	2.3 ± 0.4	2.2 ± 0.3	0.256 ^a^

^a^, Student’s *t*-test; ^b^, Mann–Whitney test.

**Table 9 ijerph-21-00338-t009:** The Poisson regression analysis for evaluating the factors independently associated with fatigue.

Variable	Prevalence Ratio (CI 95%)	*p*
Levels of trust in organizations/authorities		
Unions	0.68 (0.58–0.81)	<0.001
Workforce Management Agencies	0.64 (0.49–0.83)	<0.001
Precarious work environment and its risks		
It would be completely unacceptable.	0.71 (0.58–0.87)	<0.001

CI 95%: 95% confidence interval.

**Table 10 ijerph-21-00338-t010:** Frequencies of the words found in the port documents.

Document	Word				
	Port worker	Work	Worker	Health	Disease
1 [28]	1.020	23	00	12	01
2 [29]	1.288	88	137	99	03
3 [30]	316	04	03	09	00
4 [31]	04	03	04	00	01
5 [32]	95	24	31	25	00
Total	2.624	235	140	120	04

**Table 11 ijerph-21-00338-t011:** Categories, subcategories, and presentation of the qualitative data identified in the content analysis.

Category	Subcategory	Qualitative Data
Infrastructure and organization	Risks to human health due to expansion of the port terminal	“Poor air quality can have negative effects on human health and reduce the quality of life of port workers and the population located around the Port” (p. 46 [25])
	Lack of its own team for environmental management in the port	“In relation to port facilities [...] they do not have a specialized team to deal with environmental issues in their facilities” (p. 81 [26])
	Presence of old equipment used to carry out the work	“[…] in addition to the equipment described, the following are also found at the Port of Pelotas: a GE249 electric crane, two mobile crawler cranes and a front forklift. It should be noted that only the electric crane is in operation” (p. 189 [24]) “Lack of assessment of the situation of port equipment at the Port of Pelotas and inadequate conditions at Warehouse A3: despite the existence of port equipment at the Port of Pelotas, there is no assessment of their situations and, consequently, a defined action plan. Furthermore, according to SUPRG, Warehouse A3 needs repairs so that it can be used for cargo storage and to support operations at the Port of Pelotas” (p. 184 [25])
	Waterway access, which does not allow night navigation	“[…] there is a restriction on night navigation for vessels with a LOA of more than 111 m or those carrying dangerous cargo” (p. 59 [26])
Risk prevention	Supply of personal protective equipment (PPE)	“The port administration will provide personal protective equipment [...]. The supply is exclusive to port operations within the organized port area” (p. 55 [28])
	Regulating, controlling, and monitoring products and services, which involve a risk to public health	“The Agency is also responsible, respecting current legislation, for regulating, controlling and inspecting products and services that involve a risk to public health” (p. 28 [28])
	Care for health, hygiene, and safety standards in port work	“Item XI. ensure health, hygiene, and safety standards in port work” (p. 24 [28])

## Data Availability

The data are available upon reasonable request.

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
