# Peer review of "Risk Perception and Fatigue in Port Workers: A Pilot Study"

_ijerph, 2024, doi:10.3390/ijerph21030338_

Round 1
Reviewer 1 Report
Comments and Suggestions for Authors
There are several problems with this study:
1. The study cannot be called a mixed-methods study because the number of respondents is too low for a proper quantitative study (n=39). Authors add three references about mixed-methods (references 17-19) top write one sentence in the manuscript. I do not think followed proper methods to call the study a mixed-methods study.
2. The aims of the study seem to be a list of activities. I read the paper and I do not understand what is the main aim of the study and the justification for the study. I recommend narrowing down the conceptualization of the study. Is it the intervention that is being analyized? Was the purpose of the study the design of an intervention? Was the purpose of the study to analyze the relationship between fatigue and organizational factors? The aim is not clear.
3. The qualitative part of the study is not clear. The methods are not clearly described.
Comments on the Quality of English Language
The paper needs a full revision regarding the English language.
Author Response
Dear Editors,
The authors express their gratitude for your attention and diligence in reviewing
the manuscript 'Risk Perception and Fatigue in Port Workers: A Pilot Study.' We will now address the comments point by point.
Reviewer 1 comments - 1. The study cannot be called a mixed-methods
study because the number of respondents is too low for a proper quantitative study (n=39). Authors add three references about mixed-methods (references 17-19) top write one sentence in the manuscript. I do not think followed proper methods to call the study a mixed-methods study.
Authors' response: The authors characterized the study as descriptive and
cross-sectional, encompassing both quantitative and qualitative data.Reviewer 1 comments - The aims of the study seem to be a list of activities. I read the paper and I do not understand what is the main aim of the study and the justification for the study. I recommend narrowing down the conceptualization of the study. Is it the intervention that is being analyized? Was the purpose of the study the design of an intervention? Was the purpose of the study to analyze the relationship between fatigue and organizational factors? The aim is not clear.
Authors' response: The objectives of the study have been revised. In the
original version, the objectives were: to identify the perception of port workers
about the individual and collective risks present in the work environment; identify port workers’ trust in organizations and authorities when referring to risks associated with the work environment; assess fatigue in port workers; verify the association between fatigue and levels of trust in organizations and authorities and risk perception; analyze official documents that structure the studied port and; present health intervention with port workers.
After review, the aims of this study were to assess fatigue in port workers;
verify the association between fatigue and levels of trust in organizations, and the association between authorities and risk perception; and analyze the official
documents that structured the studied port, as well as the present health and
communication of the port workers.
Reviewer 1 comments - The qualitative part of the study is not clear. The methods are not clearly described.
Authors’ response: Following the qualitative stage, we analyzed official
documents available on the management website of the studied port. After accessing these documents, we conducted a reading and subsequent search using the following keywords: worker, port, health, disease, and risk. This approach allowed us to objectively identify how the port authorities incorporated these topics into the organization of their services. We also conducted an analysis of the frequency of the search terms. Following the initial reading, we performed a vertical analysis of the data, establishing categories and subcategories.
Reviewer 1 comments - The paper needs a full revision regarding the English
language.
Authors' response: The text was sent for review by the MDPI database.
Kind regards,
Dr. Clarice Alves Bonow
Federal University of Pelotas
Reviewer 2 Report
Comments and Suggestions for Authors
In this study from Brazil, Alves Bonow et al report on the Risk Perception on Occupational and Work-related risks and fatigue in port workers from the very south Brazilian area of Rio Grande do Sur. According to the results of this pilot study, workers reported good levels of trust in organizations/authorities, particularly for noise, fire and air pollution; moreover, around 30.8% of participants were affected by some degree of fatigue.
Unfortunately, the potential significance of these results is affected by several shortcomings. First of all, the total number of sampled workers is reduced, only 44 with results only reported from 39 port workers (in this regard, please explain how the initial sample changed from 44 to 39, I guess a dropout but it is not reported across the text). This substantial shortcoming must be addressed in the discussion section.
Second, but clearly consistent with the first point: how many workers were occupied at the Portal Authority that was involved in this study? The participation rate will be of great significance for the reader, as higher participation rates would mean a better representativity of these results. Contrariwise, this factor must be acknowledged and discussed, particularly in terms of potential self-selection of participants.
Third, the outcome of fatigue is unclearly reported. In some instances of the paper a prevalence of 38.5% is reported, then 30.8%. Please clarify.
Fourth, Figure 1, 2, 3 and 4 adds very few information to this study. Figure 1 could be removed. Figure 2 could be moved to the Appendix with a map of the Rio Grande do Sur area (please, in this regard, provide some background information on the demographic and socioeconomic features of this area). Figure 3 and 4 could be changed as graphical abstract of this study.
Author Response
Dear Editors,
The authors express their gratitude for your attention and diligence in reviewing
the manuscript 'Risk Perception and Fatigue in Port Workers: A Pilot Study.' We will now address the comments point by point.
Reviewer 2 comments - In this study from Brazil, Alves Bonow et al report on the
Risk Perception on Occupational and Work-related risks and fatigue in port workers from the very south Brazilian area of Rio Grande do Sur. According to the results of this pilot study, workers reported good levels of trust in organizations/authorities, particularly for noise, fire and air pollution; moreover, around 30.8% of participants were affected by some degree of fatigue. Unfortunately, the potential significance of these results is affected by several shortcomings. First of all, the total number of sampled workers is reduced, only 44 with results only reported from 39 port workers (in this regard, please explain how the initial sample changed from 44 to 39, I guess a dropout but it is not
reported across the text). This substantial shortcoming must be addressed in the
discussion section.
Authors' response: In line 153, there is a reference to the 5 workers who
declined to participate in the study, leading to a total of 39 participants. The sentence has been rephrased for improved clarity. Additionally, it is important to note that the study was conducted with the population of workers at the Port of Pelotas. While there were refusals, everyone had the opportunity to participate.
Reviewer 2 comments - Second, but clearly consistent with the first point: how
many workers were occupied at the Portal Authority that was involved in this study? The participation rate will be of great significance for the reader, as higher participation rates would mean a better representativity of these results. Contrariwise, this factor must be acknowledged and discussed, particularly in terms of potential self-selection of participants.
Authors' response: The participation rate suggested by Reviewer 2 has been
included. The study's participation rate was 88.6%.
Reviewer 2 comments - Third, the outcome of fatigue is unclearly reported. In
some instances of the paper a prevalence of 38.5% is reported, then 30.8%. Please clarify.
Authors' response - The prevalence of fatigue has been reviewed and corrected.
Reviewer 2 comments - Fourth, Figure 1, 2, 3 and 4 adds very few information to
this study. Figure 1 could be removed. Figure 2 could be moved to the Appendix with a map of the Rio Grande do Sur area (please, in this regard, provide some background information on the demographic and socioeconomic features of this area). Figure 3 and 4 could be changed as graphical abstract of this study.
Authors' response: Figure 1 has been removed. Figure 2 was also removed
and inserted as supplementary material, along with demographic and socioeconomic characteristics of the state of Rio Grande do Sul and Pelotas, the
municipality in which the port is located.
Figures 3 and 4 were the infographics presented to port workers. Therefore,
the authors understand that they must be presented as they are, as these are the results of the qualitative analysis that led to the graphic representation
(infographics) created to inform port workers about the research findings,
specifically regarding fatigue (Figure 3), and to present ways to prevent fatigue at work (Figure 4). The infographics were sent to the institutional emails of the 44 port workers, inquiring whether the instructions could be followed within the work environment. However, there was no response from them regarding the material sent, as workers lost access to institutional emails after leaving the port during the pandemic.
The authors are grateful for the meticulous review of the text and intend to publish it in the special issue titled 'Socio-Environmental Health and Risk Perception.
Kind regards,
Dr. Clarice Alves Bonow
Federal University of Pelotas
Round 2
Reviewer 1 Report
Comments and Suggestions for Authors
General Comments: This study aimed to analyze fatigue in port workers, specifically it aims to assess the relationship between fatigue and levels of trust in organizations, assess the relationship between authorities and risk perceptions and analyze the content of port documents. Authors have collected interesting data to which the propose to study in a quantitative manner. They had 39 participants responding to a survey containing several scales and then proceeded to do statistical analyses and made several conclusions with the findings. It is recommended that when doing statistical analyses, the sample size be at least 50 participants for chi-square analyses, their sample size is 39. This means that the findings are not very reliable or generalizable, however the data they collected is still valuable. Instead of focusing so much on the statistical analyses, authors should report their findings in a descriptive way and not jump to conclusions using the statistical analyses. They performed a review of the literature related to ports which is very important, yet they only report these findings on Table 11 and do not discuss them much. I skimmed some of the reports they reviewed and there is a lot of material there to do a deep description of the organizational measures that ports take to protect and promote workers’ health. The survey results should be descriptive and discussed, but should not be useful for statistical methods. Unless authors find
Introduction:
Paragraph 1, Lines 44-58: For the introduction, authors focus on the port industry and not on the workers, even though the study is focused on workers’ fatigue and risks. For example, authors start by specifying that ports face new global needs, but do not tell us what these global needs are. Then authors cite a study where training for port workers increased safety and produced port growth (lines 45-47). Nevertheless, there is a body of literature on how only training workers, or putting the responsibility on workers is not sustainable through time but rather there should be organizational changes that enable port growth but also and mainly workers’ safety and well-being. I suggest authors to skip lines 44-58 and focus on port workers. The study already focuses on the important part: workers, there is no need to add information about economic growth of ports. It is more relevant if authors tell us some statistics on the number of new ports around the world or the increasing number of workers in ports and the importance on focusing to protect them. Also it would be useful to provide information about any accidents or work-related prevalence of morbidity in this population to give readers an idea of the relationship between port work and port risks. Authors do that later in the introduction, but maybe there needs to be more emphasis right at the beginning of the introduction.
Lines 97-100: Please state clearly the objectives. The objectives are not as clearly delineated as in the abstract. The use of verify is not correct. “Verify” is used when researchers already know there is an association. In this case researchers want to know if there is association. Appropriate verbs would include: analyzed, assessed. For example, we assessed the relationship between fatigue and levels of trust in organizations and authorities, we assessed the relationship between fatigue and risk perceptions. Then authors did a review of the literature. They reviewed official documents. The correct verb to use is reviewed official documents, then they need to specify if these documents are government or organizational documents or both.
Materials and Methods:
Lines 105-106: This is the first time that authors state “this is a pilot study”. This information should be introduced earlier in the paper It should be introduced in the abstract and in the introduction. Then authors also mention “extreme south”, it should be “in the south of Brazil”.
Lines 105-115: This section should be the last in the methods section, but prior to it, authors should specify that one of the original aims of the study was to interview workers and due COVID they couldn’t. Also please provide more information about workers being dismissed. It is 2024 now, are workers back? Were the infographics used by the port company?
Lines 137-148: Authors mention a quantitative stage and a qualitative stage of the research. This terminology is not used in these types of studies. First of all authors had a research question and they used survey data and a review of the literature to answer the questions. I suggest that in this section authors focus only on the participants and this section would be more appropriate in the results section describing sociodemographic characteristics.
Sub-section “Questionnaires and data collection” lines 152-182: I really like how this section describes both of the fatigue scales used in this study and the content of each survey. What is missing from this section is the description of the precarious work environment survey and how participants were recruited in the study. Some context would help understand readers where participants came from. For example, is this pilot study part of a bigger study? How were researchers able to approach the port workers? Was it mandatory that workers fill our the survey during the arranged shift? Were they offered an incentive? How did they obtain the support from employers? How were the surveys administered: online, paper, interview? Is it only one port company or several?
Somewhere in methods, please include that part of the study plan was to design infographics aimed for workers. Were they also aim for managers and employers?
Results section
Line 225: Tell us the minimum wage on both the national currency and USD currency for international comparisons.
Line 225: “Average study time”, do you mean average years of education? It is not clear
Table 1: Race (Caucasian) please explain this race in the Brazilian context. “Operating time in port” should be “Years working doing port work”, it would be useful to include minimum number of year and maximum as the sample size is very low.
Lines 240-243: Please explain context of all the entities: port guard, unions, customs brokers and labor management bodies. What is the port guard? How are they linked to port workers? Are all workers in a union? What are customs brokers and how do they interact with port workers? Who is part of labor management bodies and what is their role with respect to port work?
Lines 251-254: This is the first time authors mention “precarious work environment”, they did not explain in the methods section what is a precarious work environment and how it is measured. In Occupational Health precarious work environment needs to be defined since it is a construct that is multifactorial and can be used for employment conditions or working conditions. Did authors used a validated measure of precarious work environment?
Table 5: Here it is the first time authors mention “People in general”. Are they comparing answers from port workers to the general population? Where did the information from the general population come from? Authors did not describe these details in the methods section, they should include information of these data, how they obtained it and what is the purpose of comparison with port workers. This should be done in the methods section.
Table 10: This is unnecessary information as the words are extremely general and not specific to the risks that port workers face.
Lines 375-381: Researchers designed infographics that were meant for workers, but the infographics never reached the intended audience, have researchers used them for the new workers at the port? Has the port been permanently closed?
Discussion
I have several comments regarding the discussion. I believe authors are not taking into account the rainbow model of the determinants of health (Dahlgren and Whitehead 1991, 2007) in discussing their findings which risks doing a disservice to participants of the study and the readers of the paper. The rainbow model of the determinants of health highlights that individuals are embedded in societies (e.g. jobs) with certain infrastructures and that individuals are exposed to these infrastructures in unequal ways and that it is not necessarily the fault or the decision of individuals themselves to be exposed to situations that place them at risk of higher probability to be sick. I say this for many reasons:
In Lines 398-401: Authors make assumptions about workers based on their years of education implying that not having enough years of education is a source of not being trained to do their job in a safe way or not being able to read the port documents. Here the responsibility is being placed on workers and not on the companies that hire them.
Second, the discussion is meant to compare findings with the literature both supporting evidence but also counter-evidence as well as limitations of the study. Authors do not mention the strengths nor the limitations of the study. I see one big limitation: the number of workers who answered the survey. Authors made statistical analyses with a very small sample. In a discussion section authors should approach the sources of bias in their results, especially with the use of such a small sample. Another limitation, may be the environment in which workers filled out the survey which is not completely explained in the methods section. Were they required by employers to fill out the survey? Were they informed that the survey was going to be anonymous? There could be a response bias if they think that employers will see their results. Please also discuss how generalizable are your findings and what are possible implications for research, policy and practice.
Specific comments:
Lines 410-416: This part is not clear. It is not clear how suggesting to do the work only during the day shift justifies the greater trust in managers. Please explain.
Lines 441-443: Please provide examples of the psychological aspects to which port workers are exposed to.
Comments on the Quality of English Language
The use of the word “verify” is not concordant with the objectives of the research. Please check grammar of some sentences, especially the ones explaining the findings.
Author Response
Dear Editors,
The authors express their gratitude for your attention and diligence in reviewing the manuscript 'Risk Perception and Fatigue in Port Workers: A Pilot Study.' We will now address the comments point by point.
Reviewer 1 - General Comments: This study aimed to analyze fatigue in port workers, specifically it aims to assess the relationship between fatigue and levels of trust in organizations, assess the relationship between authorities and risk perceptions and analyze the content of port documents. Authors have collected interesting data to which the propose to study in a quantitative manner. They had 39 participants responding to a survey containing several scales and then proceeded to do statistical analyses and made several conclusions with the findings. It is recommended that when doing statistical analyses, the sample size be at least 50 participants for chi-square analyses, their sample size is 39. This means that the findings are not very reliable or generalizable, however the data they collected is still valuable. Instead of focusing so much on the statistical analyses, authors should report their findings in a descriptive way and not jump to conclusions using the statistical analyses. They performed a review of the literature related to ports which is very important, yet they only report these findings on Table 11 and do not discuss them much. I skimmed some of the reports they reviewed and there is a lot of material there to do a deep description of the organizational measures that ports take to protect and promote workers’ health. The survey results should be descriptive and discussed, but should not be useful for statistical methods. Unless authors find.
Authors' response: The authors thank you for your careful review.
Reviewer 1 - Introduction:
Paragraph 1, Lines 44-58: For the introduction, authors focus on the port industry and not on the workers, even though the study is focused on workers’ fatigue and risks. For example, authors start by specifying that ports face new global needs, but do not tell us what these global needs are. Then authors cite a study where training for port workers increased safety and produced port growth (lines 45-47). Nevertheless, there is a body of literature on how only training workers, or putting the responsibility on workers is not sustainable through time but rather there should be organizational changes that enable port growth but also and mainly workers’ safety and well-being. I suggest authors to skip lines 44-58 and focus on port workers. The study already focuses on the important part: workers, there is no need to add information about economic growth of ports. It is more relevant if authors tell us some statistics on the number of new ports around the world or the increasing number of workers in ports and the importance on focusing to protect them. Also it would be useful to provide information about any accidents or work-related prevalence of morbidity in this population to give readers an idea of the relationship between port work and port risks. Authors do that later in the introduction, but maybe there needs to be more emphasis right at the beginning of the introduction.
Authors' response: The suggestion was accepted.
Reviewer 1: Lines 97-100: Please state clearly the objectives. The objectives are not as clearly delineated as in the abstract. The use of verify is not correct. “Verify” is used when researchers already know there is an association. In this case researchers want to know if there is association. Appropriate verbs would include: analyzed, assessed. For example, we assessed the relationship between fatigue and levels of trust in organizations and authorities, we assessed the relationship between fatigue and risk perceptions. Then authors did a review of the literature. They reviewed official documents. The correct verb to use is reviewed official documents, then they need to specify if these documents are government or organizational documents or both.
Authors' response: The last paragraph of the introduction was rewritten and the term “verify” was replaced by “analyze”, according to one of the reviewer’s suggestions.
Reviewer 1 - Materials and Methods: Lines 105-106: This is the first time that authors state “this is a pilot study”. This information should be introduced earlier in the paper It should be introduced in the abstract and in the introduction. Then authors also mention “extreme south”, it should be “in the south of Brazil”.
Authors' response: The information was added in the summary and introduction, as suggested by the reviewer.
Reviewer 1 – Lines 105-115: This section should be the last in the methods section, but prior to it, authors should specify that one of the original aims of the study was to interview workers and due COVID they couldn’t. Also please provide more information about workers being dismissed. It is 2024 now, are workers back? Were the infographics used by the port company?
Authors' response: In lines 94 – 102 the authors explain that the proposal included a study of other ports and structured and semi-structured interviews with workers; however, after the collection of quantitative data by the group of workers in January and February 2020, the COVID-19 pandemic began. In this context, face-to-face activities were suspended, making it necessary to reorganize activities remotely. With the advancement of the pandemic and the difficulties in returning to face-to-face activities, the workers interviewed were fired, losing their connection with the institution and access to the institutional email where they received messages from the research team. Therefore, the continuation of the project was unfeasible.
Reviewer 1 - Lines 137-148: Authors mention a quantitative stage and a qualitative stage of the research. This terminology is not used in these types of studies. First of all authors had a research question and they used survey data and a review of the literature to answer the questions. I suggest that in this section authors focus only on the participants and this section would be more appropriate in the results section describing sociodemographic characteristics.
Authors’ response: The beginning of the paragraph has been rewritten for better understanding.
Reviewer 1 - Sub-section “Questionnaires and data collection” lines 152-182: I really like how this section describes both of the fatigue scales used in this study and the content of each survey. What is missing from this section is the description of the precarious work environment survey and how participants were recruited in the study. Some context would help understand readers where participants came from. For example, is this pilot study part of a bigger study? How were researchers able to approach the port workers? Was it mandatory that workers fill our the survey during the arranged shift? Were they offered an incentive? How did they obtain the support from employers? How were the surveys administered: online, paper, interview? Is it only one port company or several?
Authors' response: As explained in the manuscript, all workers at a port were invited to participate in the study. Participation in the research was not mandatory, as there were refusals.
Reviewer 1 - Somewhere in methods, please include that part of the study plan was to design infographics aimed for workers. Were they also aim for managers and employers?
Authors' response: The main objective was the workers. Management would also have been included if there had not been the pandemic.
Reviewer 1 - Results section
Line 225: Tell us the minimum wage on both the national currency and USD currency for international comparisons.
Authors' response: The information has been added.
Reviewer 1 – Line 225: “Average study time”, do you mean average years of education? It is not clear
Authors' response – The information has been corrected.
Reviewer 1 - Table 1: Race (Caucasian) please explain this race in the Brazilian context. “Operating time in port” should be “Years working doing port work”, it would be useful to include minimum number of year and maximum as the sample size is very low.
Authors' response: The term Caucasian was changed to White. The remaining terms have been corrected.
Reviewer 1 - Lines 240-243: Please explain context of all the entities: port guard, unions, customs brokers and labor management bodies. What is the port guard? How are they linked to port workers? Are all workers in a union? What are customs brokers and how do they interact with port workers? Who is part of labor management bodies and what is their role with respect to port work?
Authors' response: Terms have been clarified.
Reviewer 1 - Lines 251-254: This is the first time authors mention “precarious work environment”, they did not explain in the methods section what is a precarious work environment and how it is measured. In Occupational Health precarious work environment needs to be defined since it is a construct that is multifactorial and can be used for employment conditions or working conditions. Did authors used a validated measure of precarious work environment?
Authors' response: No measure of precarious work was used.
Reviewer 1 - Table 5: Here it is the first time authors mention “People in general”. Are they comparing answers from port workers to the general population? Where did the information from the general population come from? Authors did not describe these details in the methods section, they should include information of these data, how they obtained it and what is the purpose of comparison with port workers. This should be done in the methods section.
Authors' response: The mention as “people in general” refers to the port worker's understanding of the exposure of people in general.
Reviewer 1 - Table 10: This is unnecessary information as the words are extremely general and not specific to the risks that port workers face.
Authors' response: The authors chose to maintain table 10, as this was the strategy to explain that some documents are not concerned with the health or risks to which port workers are linked.
Reviewer 1 - Lines 375-381: Researchers designed infographics that were meant for workers, but the infographics never reached the intended audience, have researchers used them for the new workers at the port? Has the port been permanently closed?
Authors' response: Line 358-362 - The infographics were sent to the institutional email addresses of the 44 port workers, asking them whether the instructions could be followed within the work environment. However, there were no responses from them regarding the materials sent as the workers lost access to their institutional email addresses after the port was shut down during the pandemic.
The workers received, but there was no feedback or continuity of the research due to the pandemic. The port was not closed, but research was not continued after this period.
Reviewer 1 – Discussion - I have several comments regarding the discussion. I believe authors are not taking into account the rainbow model of the determinants of health (Dahlgren and Whitehead 1991, 2007) in discussing their findings which risks doing a disservice to participants of the study and the readers of the paper. The rainbow model of the determinants of health highlights that individuals are embedded in societies (e.g. jobs) with certain infrastructures and that individuals are exposed to these infrastructures in unequal ways and that it is not necessarily the fault or the decision of individuals themselves to be exposed to situations that place them at risk of higher probability to be sick. I say this for many reasons:
In Lines 398-401: Authors make assumptions about workers based on their years of education implying that not having enough years of education is a source of not being trained to do their job in a safe way or not being able to read the port documents. Here the responsibility is being placed on workers and not on the companies that hire them.
Second, the discussion is meant to compare findings with the literature both supporting evidence but also counter-evidence as well as limitations of the study. Authors do not mention the strengths nor the limitations of the study. I see one big limitation: the number of workers who answered the survey. Authors made statistical analyses with a very small sample. In a discussion section authors should approach the sources of bias in their results, especially with the use of such a small sample. Another limitation, may be the environment in which workers filled out the survey which is not completely explained in the methods section. Were they required by employers to fill out the survey? Were they informed that the survey was going to be anonymous? There could be a response bias if they think that employers will see their results. Please also discuss how generalizable are your findings and what are possible implications for research, policy and practice.
Authors' response: The authors agree that the sample is small, however, it represents the entire population of a given port. The survey was anonymous and workers were informed about it.
Reviewer 1 - Specific comments: Lines 410-416: This part is not clear. It is not clear how suggesting to do the work only during the day shift justifies the greater trust in managers. Please explain.
Authors' response: Working a shift would decrease fatigue, which could contribute to confidence.
Reviewer 1 - Lines 441-443: Please provide examples of the psychological aspects to which port workers are exposed to.
Authors' response: Mental fatigue.
Reviewer 1 - Comments on the Quality of English Language - The use of the word “verify” is not concordant with the objectives of the research. Please check grammar of some sentences, especially the ones explaining the findings.
Authors' response: The word “verify” was replaced in the objectives by “analyze”, as suggested by the reviewer. The text was submitted to English language assessment at MDPI.
Reviewer 2 - Estimated Authors,
I've appreciated the efforts you paid in order to improve the overall quality of the present paper.
You did reply in a very appropriate way to all my concerns and requests, and therefore I've no further methodological issue to raise.
From a formal point of view, the paper is now nearly ready to be accepted for publication.
Comments on the Quality of English Language
Unfortunately, the overall quality of the English text remains affected by several typos. Authors are requested to perform a further effort to improve the main text.
Authors' response: We thank you for your careful review and inform you that the text was submitted for English evaluation by MDPI.
The authors are grateful for the meticulous review of the text and intend to publish it in the special issue titled 'Socio-Environmental Health and Risk Perception.
Kind regards,
Dr. Clarice Alves Bonow
Federal University of Pelotas

Reviewer 2 Report
Comments and Suggestions for Authors
Estimated Authors,
I've appreciated the efforts you paid in order to improve the overall quality of the present paper.
You did reply in a very appropriate way to all my concerns and requests, and therefore I've no further methodological issue to raise.
From a formal point of view, the paper is now nearly ready to be accepted for publication.
Comments on the Quality of English Language
Unfortunately, the overall quality of the English text remains affected by several typos. Authors are requested to perform a further effort to improve the main text.
Author Response

(The authors gave the same response as above.)
